# The effect of risk framing on support for restrictive government policy regarding the COVID-19 outbreak

Kirill Chmel [1,2]*, Aigul Klimova [1,3], Nikita Savin [3,4]

**1** Ronald F. Inglehart Laboratory for Comparative Social Research, HSE University, Moscow, Russian Federation, **2** Department of Integrated Communications, Faculty of Communications, Media, and Design, HSE University, Moscow, Russian Federation, **3** Department of Sociology, Faculty of Social Sciences, HSE University, Moscow, Russian Federation, **4** Politics & Psychology Research Laboratory, HSE University, Moscow, Russian Federation

These authors contributed equally to this work.
* kchmel@hse.ru

**Data Availability Statement:** The data underlying the results presented in the study are available from Kirill Chmel's GitHub repository https://github.com/KirillChmel/covid-risk-framing.

## Abstract

This confirmatory research investigates the influence of risk framing of COVID-19 on support for restrictive government policy based on two web survey experiments in Russia. Using 2x2 factorial design, we estimated two main effects–factors of risk severity (low vs. high) and object at risk (individual losses vs. losses to others). First, focusing on higher risks had a positive effect on support for the government's restrictive policy. Second, focusing on the losses for others did not produce stronger support for the restrictive policy compared to focusing on personal losses. However, we found a positive moderation effect of such prosocial values as universalism and benevolence. We found that those with prosocial values had a stronger positive effect in the "losses for others" condition and were more willing to support government restrictive policy when others were included. The effects found in our experimental study reveal both positive and negative aspects in risk communication during the pandemic, which may have a great and long-term impact on trust, attitudes, and behavior.

## Introduction

As COVID-19 turns into a pandemic, a political debate is simultaneously raging about whether autocracies or democracies are better at fighting epidemics [1]. Media pundits and global health officials praise draconian security measures imposed by the Chinese government to prevent the spread of COVID-19 [2, 3], and severely criticize the Swedish government for being excessively lax and soft about containing the virus [4]. But how many people would rather 'stay at home' and keep a safe 'social distance' instead of reaping the benefits of limitless freedom which most of them enjoyed before the COVID-19 outbreak? As of March 2020, approximately 75% of the world population, says Gallup International Association [5]. Indeed, according to a host of public opinion polls, the percentage of those willing to sacrifice some of their human rights to stave off the spread of coronavirus varies from 32% in Japan to 95% in

**Funding:** The article was prepared within the framework of the HSE University Basic Research Program. The funders had no role in study design, data collection and analysis, decision to publish, or preparation of the manuscript.

**Competing interests:** The authors have declared that no competing interests exist.

Austria. In Russia, where the first cases had just been registered to that moment, there were 60% [5]. About a year later, in December 2020 the world on average became less willing to sacrifice rights to prevent the spread of the virus—the percentage dropped from 75% to 70%. At the same time, the statistics in Russia dramatically changed, since citizens were hardly willing to sacrifice their rights. Only 39% of Russian citizens were positive about their rights being trampled in order to resolve the public health crisis [6].

One of the possible explanations of willingness to sacrifice some human rights is rooted in the idea of risk perception as a driving force for decision-making. Research argues that risk perception drives support for security policies that infringe on civil liberties. Particularly, willingness to trade off civil liberties for security increases in the aftermath of exogenous shocks including, probably, public health outbreaks. The term, 'When in danger, turn right', from Karwowski et al. [7], demonstrates how the COVID-19 threat promotes social conservatism and support for right-wing candidates. Nonetheless, the perception of risk does not develop on its own, so media coverage, for instance, may introduce a disjuncture between perceptions of personal risk and objective estimates of population incidence [8].

Indeed, framing of a message, which is selection and emphasis on some aspects of a message, can have a greater impact on attitudes and behavior than the actual context [9]. Recent studies address the effects of different framings on following the protective measures during the pandemic. For instance, in line with the research on gain-loss framing, scholars found that gain-framed messages are more effective in promoting self-care behaviors [10–12]. At the same time, other scholars either got null results [13], or found that framing effects are observed if conditioned by such individual characteristics as political ideology or socio-demographics [14, 15].

Risk communication is critical to managing public health outbreaks because of deep uncertainty and lack of issue localization [16]. Risk messages presented to citizens openly and timely aim to rectify the knowledge gap in understanding an epidemiological crisis and adjust the public's behavior to cope with the risk proactively [17]. However, not all messages have the same effect on citizens' behavior. For instance, while the World Health Organization (WHO) proclaimed the importance of being 'supportive' and 'careful' towards others [18], the Swedish strategy to manage COVID-19 has been largely based on the personal responsibility of the citizens who receive daily information about, and individually targeted instructions for, self-protection. Which strategy—to protect yourself or to take care of others—works better then?

## Political effects of high and low risk framing

The psychometric paradigm in risk perception research suggests that the main risk categorizations are the level of dread people feel about the risk and the familiarity with the risk [19, 20]. The main factor was found to be the dread risk. The higher the score on this factor, the more individuals are willing to support any restrictive measures that can reduce this risk [19]. This paradigm emphasizes the importance of affective responses that individuals have towards potential risk. They coined the term 'affect heuristic' to describe these affective responses [20, 21]. Heuristics are biases whereby respondents use only part of the information with which they are provided [22]. Researchers found that the higher the level of dread people feel about the risks, especially if the risks are new and they are frequently discussed in media, the higher the perceived risks are [23].

Higher perceived risks are the main source of enforcing authoritarian attitudes. Social threats increase right-wing authoritarianism [24–26]. The trade-off between civil rights and a high threat to society drives up the willingness to sacrifice rights to reinforce social order [27]. This can be found in risk situations where there is a threat to some particular groups or society

overall in the aftermath of pandemics [28], violent crimes, political or economic crises [29], terrorist attacks [30], natural disasters [31], or climate change [32]. Overall, individuals tend to preserve collective security at the expense of freedom, autonomy, and rights, when there are high perceived risks to society.

We rely on the literature that emphasizes the psychological mechanisms behind framing effects and the literature which shows the effect of framing on a number of attitudes [33, 34]. Frames provide some meanings to the events, selecting certain aspects of the perceived reality and making them more salient in communication [35]. They make some opinions available for retrieval and accessible while being exposed to them. Following the distinction, which is made by Chong and Druckman [33], between issue and equivalence framing, we use the former approach to increase the ecological validity of research. We promote high-risk and low-risk as different considerations, since they are usually communicated by politicians. Despite their logical similarity they may not be necessarily perceived as a trade-off without public discussion. As a result, we expect to find that the high-risk framing of COVID-19 would increase support for restrictive government policy compared to low-risk framing.

**H1:** High-risk framing will have a stronger effect on the support for restrictive government policy compared to low-risk framing.

## Who is the main object at risk?

The question of 'object at risk' is one of the main categorizations which has an effect on risk perception [36]. COVID-19 is widely recognized to pose a threat to all age groups. However, according to statistics and media, as well as according to restrictive policy regarding particular groups of population, older people are particularly vulnerable, thus they are central to the COVID-19 risk [37]. The politicians and media justified restrictive measures by emphasizing that people themselves can be risk objects, which poses a threat to others, especially for those who are at higher risk. That produced the international 'Stay home. Save lives' media campaign. The discourse framed the restrictive COVID-19 measures as a way to save the lives of others. The WHO promoted not only protecting oneself from getting sick but others as well [18] along with being 'supportive' and 'careful' towards others.

Though economic perspective focuses on self-interest, a number of researchers stress the importance of prosocial behavior. Research shows that higher responsibility for oneself and for others regarding risky decisions, decreases risk willingness and risky behavior [38, 39]. There is evidence that people tend to be more risk averse when they make a choice for others, rather than for themselves [40]. Indicating that vaccination is a prosocial behavior, which results not only in direct protection of the vaccinated but also protection of unvaccinated individuals through herd immunity, increases the willingness to be vaccinated [41, 42]. Individuals tend to be more risk-averse and cautious when their risky behavior can have a negative effect on others [43, 44], or are more willing to take risks in order to help others [45].

In words of Mary Douglas [46], during the COVID-19 pandemic, everyone is described as potentially 'contaminated', 'polluted', and dangerous to others as everyone can be the asymptomatic carrier of COVID-19 without knowing it, as the symptoms can take up to 14 days to manifest themselves. As a result, everyone can be accused of being a threat to others, and society overall, and punished for not following the rules of self-isolation and staying at home, as breaking the rules can be dangerous to the health of others and society.

Recent research on the effect of prosocial messages on COVID-19 prevention behaviors showed some mixed evidence. While some experimental studies found almost no difference between self-focused and prosocial framing on the willingness to self-isolate and wash hands

more often [47, 48], other studies found that prosocial framing indeed increased such preven-tion behavior as social distancing [49–51] and wearing masks [52]. Some other experimental studies found no effect of prosocial framing on clickthrough rates while delivering advertise-ments on Facebook, compared to self-focused framing [53] and comprehension of the infor-mation regarding recommended behaviors [54]. In spite of this mixed evidence, we suggest that the framing which highlights losses to others would increase support for restrictive gov-ernment policy compared to the condition in which we highlight only personal losses.

**H2:** The object at risk which focuses on losses to others will have a stronger effect on the sup-port for restrictive government policy compared to the personal losses framing.

## Prosocial values as moderators of framing effects

Following the argument made by Chong and Druckman [33], we suggest that the effect of frames on attitudes is moderated by personal values. We assumed that the frame affects only particular types of individuals with strong prosocial attitudes. It was found in previous litera-ture that such variables as values [55, 56] or personality traits [57] could explain the variation in attitudes towards prosocial behavior like moral decision-making or altruism. According to the norm activation theory, different frames would activate particular personal values depend-ing on the individual's cognitive structure of these values [55]. As a result, the 'losses to others' frame may have an effect only if this is in line with personal values. Some studies showed that higher empathy increases willingness to self-isolate and maintain social distancing during the COVID-19 pandemic [58, 59].

We would apply the Schwartz theory of basic human values and his approach to the mea-surement of values [60]. Schwartz suggested that there are ten basic human values across cul-tures. Two of them are in the self-transcendence direction (i.e., prosocial values): benevolence and universalism [61]. Both values suggest that individuals are concerned with the welfare and interests of others, which basically means the transcendence of selfish interests. While benevo-lence enhances the welfare of in-group members, universalism enhances the welfare of all peo-ple beyond the in-group. This theory and approach to the measurement of values showed quite a high validity across many cultures and is used in such cross-cultural studies as the European Social Survey [62]. Some authors argue that such self-transcendent values should have a positive effect on a compliance with COVID-19 restrictive government policy though no empirical evidence has been shown [63]. The mechanism behind it is based on the prioriti-zation of the interests of others at some personal cost. Therefore, we suggest that the 'losses to others' framing has a stronger effect on individuals with prosocial values, i.e. with stronger benevolence and universalism values.

**H2a:** The effect of the object at risk, which focuses on losses to others, will have a stronger effect on the support for restrictive government policy compared to the personal losses framing for the individuals with strong prosocial attitudes.

## The context of the study: COVID-19 in Russia

Two experiments were conducted in Russia. Experiment 1 was conducted during the so-called first wave of COVID-19. On March 28, 2020 –the day when Experiment 1 started–the con-firmed number of cases in Russia was 1,264. Four people had died from COVID-19 up to that day. On April 24, 2020 when Experiment 1 was finished, a total of 68,622 cases were con-firmed, and the COVID-19 death toll reached 615. On March 25, 2020 the President, Vladimir Putin, declared a non-working week in all Russian regions from March 28 to April 5, 2020

[64], which was later extended till April 30 [65], and then till May 11, 2020 [66]. He also entrusted regional authorities full powers to adopt restrictive measures depending on the number of cases in a region [65]. For instance, Moscow, Saint-Petersburg and a number of other Russian regions were placed under lockdown on March 30, 2020 by the decision of local authorities. People were allowed to go outside for medical care purposes, shopping for food and medication, and going to work if remote work was not an option. Besides the obligatory closure of schools, universities, gyms, swimming pools, shopping malls and hair salons, a number of regions including Moscow and Saint-Petersburg launched a digital pass system in April to allow residents to leave their homes for essential reasons as well as a smartphone app to monitor coronavirus patients' movement in self-isolation [67].

Experiment 2 was conducted during the so-called second wave of COVID-19. As compared with the first wave, the situation with COVID-19 in Russia had largely changed. On November 13, 2020 when Experiment 2 started, the confirmed number of cases in Russia was 1,880,551 and the COVID-19 death toll reached 32,443. During Experiment 2 no lockdown was in place in Russia, as well as there were no non-working weeks. The system of digital passes, which was abolished in all Russian regions by June 9, was no longer used. However, in November 2020 a series of restrictive measures, which were lifted in summer, had been reintroduced across Russian regions to contain the spread of the coronavirus. Nevertheless, such measures as mandatory wearing of face masks and gloves, prohibition of mass events and mass gatherings, reduced capacity of theaters, cinemas, and restaurants, and maintaining social distance had never been lifted from the times of the first wave. The head of Russian Federal Service for Surveillance on Consumer Rights Protection and Human Wellbeing (Rospotrebnadzor), Anna Popova, called for restrictive measures in the regions with the highest numbers of the COVID-19 active cases; and Russian Prime Minister, Mikhail Mishustin, supported this idea [68]. Following this order, for instance, in Moscow, such measures as distance learning and remote work were introduced.

By November 2020 two vaccines against COVID-19 were officially registered in Russia. On 11 August 2020, Russian president Vladimir Putin announced the official approval of the Sputnik-V vaccine. Two month later, on 14 October 2020 another vaccine, EpiVacCorona, was officially registered. However, according to Russian public opinion polls, the vaccine did not make citizens less fearful of the virus. While about 57% of Russians were afraid of getting sick with COVID-19 during the first wave in March 2020, in October 2020 the percentage increased up to 64% [69].

## Experiment 1

### Participants

Current undergraduate or graduate students of the HSE University were eligible to take part in the experiment, except for students of Political Science and Sociology departments. The consent was obtained in a written form. We invited students via their group emails, which are used for communication between lecturers and students. As a result, according to AAPOR [70] standards, response rates cannot be computed. The number of completed interviews was 762 (N = 762). Completed interviews were determined as those which had more than 80% of the essential questions answered [70].

No course credits were provided for the survey participation and students signed a consent form in which they were told they were free to withdraw from participation at any time they wanted. As an incentive for survey completion, we offered participation in a lottery in which students could win the smart home device, Yandex. Station (the price of around $150 U.S.). The break-off rate was 46% (N = 760), about half of the breakoffs were at the introduction

page. Some individuals who reported that they are not students of the HSE University were screened out (N = 118). On average, it took 23 minutes to complete the survey (M = 23.12, SD = 12.69). The baseline characteristics of the final sample are given in S1 File.

Although we used a convenience sample, the use of such samples does not appear to consistently generate false negatives, false positives, or inaccurate effect sizes [71]. Kühberger [34] also finds that the behavior of student participants does not significantly differ from the behavior of non-student participants. However, we cannot estimate conditional average treatment effects (CATEs) using a convenience sample of students for the following reason. We did not expect to observe large variation in values in a convenience sample; there is some evidence that the differences in prosocial values among students are relatively small [72]. Since Mullinix et al. [71] suggest that estimation of CATEs in the experimental studies is problematic when there is lack of variance of the moderator, especially among convenience samples of students, we did not test hypothesis H2a in Experiment 1.

## Experimental design

The experimental design was approved by the Council of Peers at Ronald F. Inglehart Laboratory for Comparative Social Research (№ER-2020-01). It is confirmed that the proposed research project conforms to ethical standards in modern social sciences. We designed a web-based experiment with a 2 (risk severity: high vs. low) X 2 (object at risk: individual losses vs. losses to others) factorial design. The subjects were randomly assigned to one of the four conditions: (1) High-risk X Individual losses, (2) High-risk X Losses to others, (3) Low-risk X Individual losses, (4) High-risk X Losses to others. The randomization process was carried out at the individual level and was conducted within the flow of the survey. No restrictions were placed on randomization, and we did not employ blocking. To ensure the effectiveness of randomization we checked for the covariate balance and put these results in S2 File. The p-values of joint orthogonality tests indicate that the group differences are insignificant, except for the probability of COVID-19 infection in the manipulation of the risk severity factor (for the details of statistical analysis, see S2 File). The data collection started on March 28, 2020 and finished on April 24, 2020. As is recommended in Reporting Guidelines for Experimental Research, the CONSORT flow diagram is provided in S3 File.

## Materials

The vignettes were structured as the set of rubrics with essentially similar content, yet different framing. The information on the pandemic, scale of the issue, probability of infection / recovery, medical treatment, long-term negative consequences for personal health (yes / no), incubation period, and advice on how to protect yourself / take care of others, were described in texts. The statistics and information in the vignettes were taken from official websites such as the World Health Organization, Russian Federal Service for Surveillance on Consumer Rights Protection and Human Wellbeing, and Russian media sources, in March-April 2020. The length of text varied from 214 to 265 words. Vignettes can be found in Table 1. To ensure that the independent variable had effectively been manipulated and the participants understood risk framing in the way we wanted them to, we used three manipulation checks. Further details of manipulation checks are given in S8 File.

## Outcome measures and covariates

We had three dependent variables mainly used to ensure the robustness. Descriptive statistics of the dependent variable measures and pre-treatment covariates, which are described in S4 File, are given in Table 2. First, we asked the participants 'How willing are you to sacrifice

some of your rights if this helps prevent the spread of coronavirus in Russia?', and measured their willingness to sacrifice rights on a 5-point scale (1 –'not at all willing'; 5 –'fully willing').

Second, we provided the participants with a list of 22 restrictive government policies which either had already been adopted by the Russian government or were being discussed in order to prevent the further spread of COVID-19. The complete list of security measures is given in

**Table 1. Experimental vignettes used in Experiment 1.**

| | Individual Losses | Losses to Others |
|---|---|---|
| **Low Risks** | **Flu pandemic** | **Flu pandemic** |
| | • World Health Organization (WHO) has announced an outbreak of a novel coronavirus influenza pandemic. An influenza pandemic is announced when a new influenza virus appears and spreads around the world. | • World Health Organization (WHO) has announced an outbreak of a novel coronavirus influenza pandemic. An influenza pandemic is announced when a new influenza virus appears and spreads around the world. |
| | **Spread** | **Spread** |
| | • World Health Organization recognizes that for most people the risk of infection with a novel coronavirus is very low. Seasonal flu still remains the most common respiratory disease which every year kills up to 650,000 people worldwide. | • World Health Organization recognizes that for most people the risk of infection with a novel coronavirus is very low. Seasonal flu still remains the most common respiratory disease which every year kills up to 650,000 people worldwide. |
| | **Recovery rates** | **Recovery rates** |
| | • On average, 96 out of 100 people recover from the novel coronavirus.<br>• Countries that have made great efforts to track and trace infected people show that 99 out of 100 people recover—statistics similar to the seasonal flu. | • On average, 96 out of 100 people recover from the novel coronavirus.<br>• Countries that have made great efforts to track and trace infected people show that 99 out of 100 people recover—statistics similar to the seasonal flu. |
| | **Health risks** | **Health risks** |
| | • In most cases, the symptoms are mild, so no specific medical treatment is required. | • In most cases, the symptoms are mild, so no specific medical treatment is required. |
| | **Incubation period** | **Effective medications** |
| | • A person can become infected with the novel coronavirus, but it can take up to 14 days for symptoms to appear. Due to this a person can infect other people without knowing that he can be dangerous to others. | • Some antiviral drugs such as Favipiravir have been found to be effective in treating the coronavirus. In Russia, Favipiravir will be available soon. |
| | **Effective medications** | **Why we can be dangerous to others?** |
| | • Some antiviral drugs such as Favipiravir have been found to be effective in treating the coronavirus. In Russia, Favipiravir will be available soon. | • A person can become infected with the novel coronavirus, but it can take up to 14 days for symptoms to appear. Due to this a person can infect other people without knowing that they can be dangerous to others. So, one infected individual can infect about 5 other people, which allows the disease to spread rapidly and increase the number of infected exponentially. |
| | **How can I protect myself?** | **We should consider the risks of others** |
| | • Due to the fact that many people are at high risk of death and negative health consequences if they become infected with a novel coronavirus, the World Health Organization recommends a series of measures to protect your own health. The most important and primary measure is regular and thorough handwashing, as well as compliance with the rules of respiratory hygiene.<br>In addition, tough measures against the spread of the virus are aimed at reducing the spread of infection. | In most cases, even no specific medical treatment is required to recover from a novel coronavirus. However, despite the fact that the disease often proceeds in a mild form, the World Health Organization suggests taking care not only of yourself, but also of other people. We can impact not only our health, but also the health of other people. Thus, the spread of the novel coronavirus depends on the actions of each of us. Tough measures against the spread of the virus, if each of us follows them, are aimed at reducing the spread of the infection. |

(*Continued*)

**Table 1.** (Continued)

|  | **Individual Losses** | **Losses to Others** |
|---|---|---|
| **High Risks** | **Pandemic** | **Pandemic** |
|  | • World Health Organization (WHO) has announced an outbreak of a novel coronavirus infection by pandemic. A pandemic is a global outbreak. People in more than 150 countries were infected with a novel coronavirus. | • World Health Organization (WHO) has announced an outbreak of a novel coronavirus infection by pandemic. A pandemic is a global outbreak. People in more than 150 countries were infected with a novel coronavirus. |
|  | **Spread** | **Spread** |
|  | • The WHO experts estimate that up to two-thirds of the world's population can be infected by the novel coronavirus, which means that up to 5 billion people can be infected. With the current mortality rate that means up to 200 million people can die from the novel coronavirus. | • The WHO experts estimate that up to two-thirds of the world's population can be infected by the novel coronavirus, which means that up to 5 billion people can be infected. With the current mortality rate that means up to 200 million people can die from the novel coronavirus. |
|  | **Mortality rate** | **Mortality rate** |
|  | • On average 4 out of 100 infected people are killed by the novel coronavirus.<br>• There is a risk of severe form of disease and serious health consequences.<br>• One out of five infected people experiences severe symptoms of the disease.<br>• There is a potential decrease in lung function by 20–30% even after recovery. Thus, lung problems may persist after recovery. | • On average 4 out of 100 infected people are killed by the novel coronavirus.<br>• There is a risk of severe form of disease and serious health consequences.<br>• One out of five infected people experiences severe symptoms of the disease.<br>• There is a potential decrease in lung function by 20–30% even after recovery. Thus, lung problems may persist after recovery. |
|  | **Incubation period** | **Effective medications and vaccines** |
|  | • A person can become infected with the novel coronavirus, but it can take up to 14 days for symptoms to appear. Due to this a person can infect other people without knowing that he can be dangerous to others. | • There is currently no known medication proven to treat the disease nor the vaccine. |
|  | **Effective medications and vaccines** | **Why we can be dangerous to others?** |
|  | • There is currently no known medication proven to treat the disease nor the vaccine. | • A person can become infected with the novel coronavirus, but it can take up to 14 days for symptoms to appear. Due to this a person can infect other people without knowing that they can be dangerous to others. So, one infected individual can infect about 5 other people, which allows the disease to spread rapidly and increase the number of infected exponentially. |
|  | **How can I protect myself?** | **We should take into consideration the risks of others** |
|  | • Due to the fact that many people are at high risk of death and negative health consequences if they become infected with a novel coronavirus, the World Health Organization recommends a series of measures to protect your own health. The most important and primary measure is regular and thorough handwashing, as well as compliance with the rules of respiratory hygiene.<br>In addition, tough measures against the spread of the virus are aimed at reducing the spread of infection. | Due to the fact that:<br>• Many people are at particularly high risk of death if they are infected with a novel coronavirus<br>• The infection spreads very quickly<br>The World Health Organization suggests taking care not only of yourself, but also of other people. We can impact not only our health, but also the health of other people. Thus, the spread of the novel coronavirus depends on the actions of each of us. Tough measures against the spread of the virus, if each of us follows them, are aimed at reducing the spread of the infection. |

S5 File. So, we asked the participants 'To what extent do you support the following measures to prevent the spread of coronavirus in Russia' and measured their level of support on a 5-point scale from 1 –'do not support at all' to 5 –'fully support'. We then added up the points

**Table 2. Descriptive statistics of dependent variables and pre-treatment covariates.**

| Variable | N | M / % | SD | Min | Max |
|---|---|---|---|---|---|
| *Dependent Variables*: | | | | | |
| Willingness to sacrifice rights | 762 | 3.34 | 0.94 | 1 | 5 |
| Support for restrictive government policy | 762 | 93.05 | 13.08 | 38 | 110 |
| Support for criminal liability for quarantine violation | 762 | 2.87 | 1.33 | 1 | 5 |
| *Control Variables*: | | | | | |
| Female* | 729 | 77% | -- | -- | -- |
| Have relatives older than 60* | 729 | 18% | -- | -- | -- |
| Probability of COVID-19 infection | 747 | 31.44 | 26.17 | 0.00 | 100.00 |
| Scale of COVID-19 in Russia | 762 | 2.99 | 0.86 | 1 | 4 |
| Frequency of check-ups | 729 | 3.12 | 0.96 | 1 | 5 |
| Government capacity to deal with the pandemic | 758 | 3.77 | 0.95 | 1 | 5 |
| Watching pro-government news | 747 | 2.05 | 1.56 | 1 | 6 |

*Note*: Dummy variables are marked with an asterisk.

from respondents' answers and used the resulting sum as the measure of participants' support for restrictive government policy (Cronbach's alpha, α = 0.91; [0.90; 0.92]).

Third, we asked the participants 'To what extent do you support the introduction of criminal liability for violation of the quarantine in Russia?' and measured their support for criminal liability on a 5-point scale (1 –'do not support at all'; 5 –'fully support'). This question appeared separately from other questions on restrictive government policies, since the intention of the Russian government to adopt it was harshly criticized in various media sources. Many people found this measure too severe and violating human rights, which resulted in public outrage on social media.

## Results

Group means comparisons are summarized in Table 3 and Fig 1. As expected, we found statistically significant differences in the support for restrictive government policy (F(3, 762) = 4.68, $p < 0.01$) and support for criminal liability for the quarantine violation (F(3, 762) = 2.72, $p < 0.05$) between four treatment conditions. Though we did not find evidence of significant differences in means of the willingness to sacrifice rights (F(3, 762) = 1.82, $p = 0.142$) between experimental conditions, the effect of risk framing was proven to be statistically significant in 2

**Table 3. Group means of experimental conditions in a completely randomized 2x2 factorial design.**

| Variable | Individual losses | | Losses to others | | *F* |
|---|---|---|---|---|---|
| | **Low-risk** | **High-risk** | **Low-risk** | **High-risk** | |
| Willingness to sacrifice rights | 3.28 (0.96) | 3.44 (0.93) | 3.23 (0.95) | 3.39 (0.92) | 1.82 |
| Support for restrictive policy | 91.57 (13.88) | 94.51 (12.65) | 90.9 (14.14) | 94.91 (11.31) | 4.68** |
| Support for criminal liability | 2.74 (1.37) | 2.96 (1.34) | 2.72 (1.31) | 3.03 (1.28) | 2.72* |
| N | 180 | 193 | 184 | 205 | |

*Note*: Group means and standard deviations (in brackets) are given in the table. F-statistics are given in the last column. Significance levels are at

*p<0.05

**p<0.01

***p<0.001.

All tests are two-tailed.

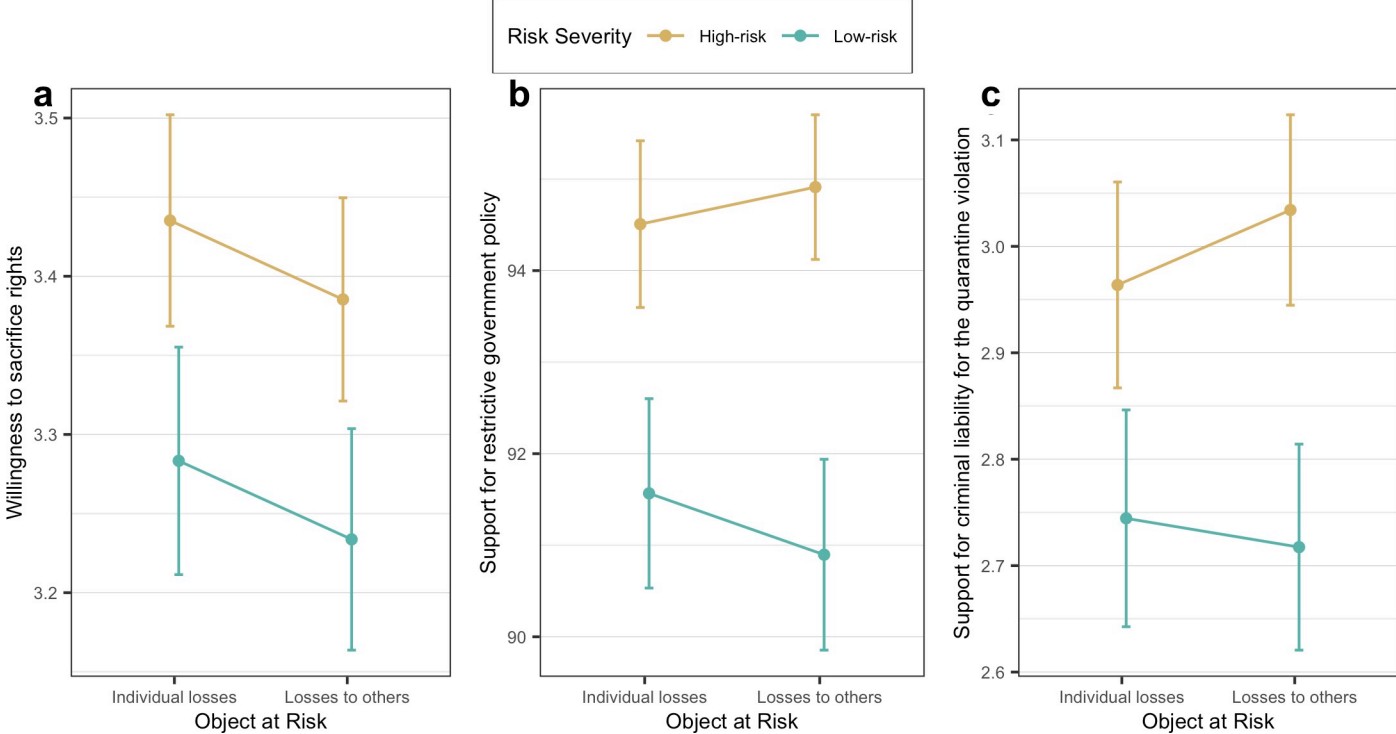

**Fig 1.** From left to right, the group means with 95% error bars in a) the willingness to sacrifice rights, b) the support for restrictive government policy, and c) the support for criminal liability for quarantine violation.

of 3 measures for the dependent variable. Pairwise comparisons of experimental conditions and be found in S6 File.

To estimate average treatment effects of the main factors, we ran t-tests. We found strong support of H1 for all three measures of the dependent variable. The differences between means of participants' willingness to sacrifice rights in high-risk (M = 3.41, SD = 0.92) and low-risk (M = 3.26, SD = 0.96) conditions are statistically significant (t(760) = 2.222, $p < 0.05$), so the ATE of risk severity on the willingness to sacrifice rights is 0.15 ([0.02; 0.29]) on a 5-point scale, Cohen's $d$ = 0.16. The same is true for the measure of support for restrictive government policy. The high-risk group mean (M = 94.72, SD = 11.97) and the low-risk group mean (M = 91.23, SD = 14) are significantly different (t(760) = 3.707, $p < 0.001$), meaning that the ATE of risk severity on the support for restrictive government policy is 3.49 ([1.64; 5.34]), Cohen's $d$ = 0.27. Regarding the support for criminal liability for the quarantine violation, we also found a statistically significant difference (t(760) = 2.805, $p < 0.01$) between the high-risk (M = 3.00, SD = 1.31) and low-risk groups (M = 2.73, SD = 1.34), so the ATE is 0.27 ([0.08; 0.46]) on a 5-point scale, Cohen's $d$ = 0.20.

OLS regression models (see Table 4; see also S7 File) confirm the results of t-tests. Overall, these results prove that the high-risk framing has a stronger effect compared to the low-risk framing on willingness to sacrifice rights (β = 0.152, $p < 0.05$), the support for restrictive measures (β = 3.489, $p < 0.01$), and the support for criminal liability for the quarantine violation (β = 0.269, $p < 0.01$).

In contrast, we did not find any evidence of H2. We found that the difference in willingness to sacrifice rights between the 'individual losses' (M = 3.36, SD = 0.95) and the 'losses to others' (M = 3.31, SD = 0.94) groups is not statistically significant (t(760) = 0.708, $p = 0.479$). There is

**Table 4. OLS regression models estimates of main effects and interactions; pre-treatment covariates as controls are not included.**

| | *Dependent variable*: | | | | | |
| | Willingness to sacrifice rights | | Support for restrictive government policy | | Support for criminal liability | |
| | (1) | (2) | (3) | (4) | (5) | (6) |
| Intercept | 3.283*** | 3.283*** | 91.283*** | 91.567*** | 2.719*** | 2.744*** |
| | (0.060) | (0.070) | (0.830) | (0.968) | (0.085) | (0.099) |
| *Factor of Risk Severity*: High-Risk | 0.152* | 0.152 | 3.489*** | 2.941* | 0.269** | 0.219 |
| | (0.068) | (0.097) | (0.942) | (1.346) | (0.096) | (0.137) |
| *Factor of Risk Target*: Losses to Others | -0.050 | -0.050 | -0.109 | -0.670 | 0.024 | -0.027 |
| | (0.068) | (0.099) | (0.941) | (1.362) | (0.096) | (0.139) |
| High-Risk X Losses to Others | | -0.0002 | | 1.074 | | 0.097 |
| | | (0.136) | | (1.885) | | (0.192) |
| N | 762 | 762 | 762 | 762 | 762 | 762 |
| Adjusted R² | 0.005 | 0.003 | 0.015 | 0.014 | 0.008 | 0.007 |
| F Statistic | 2.734† | 1.821 | 6.867** | 4.682** | 3.959* | 2.722* |

*Note*: Unstandardized beta coefficients are given in the table. Standard errors are in parentheses. Significance levels are at

†p<0.1

*p<0.05

**p<0.01

***p<0.001.

All tests are two-tailed.

also no evidence that the support for restrictive government policy is any different (t(760) = 0.080, $p$ = 0.937) for those who were exposed to personal losses framing (M = 93.09, SD = 13.32) in comparison with those who were shown the object at risk, which indicates losses to others (M = 93.01, SD = 12.87). The support for criminal liability for the quarantine violation was not proved to be appreciably different (t(760) = 0.274, $p$ = 0.784) between the 'individual losses' (M = 2.86, SD = 1.36) and the 'losses to others' (M = 2.88, SD = 1.30) conditions. Therefore, there are no grounds for accepting the second hypothesis. We conclude that the object at risk which indicates losses to others does not have a stronger effect on the support for restrictive government policy compared to the personal losses framing. OLS regression models (see Table 4; see also S7 File) confirm the results of t-tests; there are no statistically significant effects of 'losses to others' frame compared to 'individual losses' on willingness to sacrifice rights (β = -0.05 $p$ = 0.465), the support for restrictive measures (β = -0.109, $p$ = 0.908), and the support for criminal liability for the quarantine violation (β = 0.024, $p$ = 0.804).

We also estimated interaction effects between main experimental factors. Models 2, 4, and 6 in Table 4 demonstrate that there are no statistically significant interaction effects between risk severity factor and object at risk, neither on willingness to sacrifice rights (β = -0.000, $p$ = 0.999), nor on the support for restrictive government policy (β = 1.074, $p$ = 0.569), nor on the support for criminal liability for the quarantine violation (β = 0.097, $p$ = 0.612).

Interestingly enough, we found that the participants who were exposed to the 'low-risk' framing (M = 2.91, SD = 0.59) were less convinced of the credibility of the information (t(760) = -4.938, $p$ < 0.001) than those who were in the 'high-risk group' (M = 3.1, SD = 0.49) (see S8 File for more details). We found that the CATEs (conditional average treatment effects) of risk severity framing increase for those who perceived the information as credible (see S8 File).

## Experiment 2

### Participants

To calculate the CATEs and effect sizes among the general population, we conducted the second study during the so-called 'second wave of COVID-19' using a volunteer online access panel, managed by Online Market Intelligence (OMI) in Russia. The panel has ISO 20252 certification. The consent was obtained in a written form. The number of completed interviews was 1,570. The participation rate [70] was 5%. The break-off rate was 9.6% (N = 187). Some respondents were screened out (N = 7) or started completing the survey when some quotas were full (N = 181). There were some nationally representative quotas on gender, age, federal district, and level of education.

On average, it took 29 minutes to complete the survey (M = 28.6; SD = 20.41). We have excluded from the analysis those respondents who showed low data quality, which is extremely quick reading of the vignettes and straight lining in grid questions [73]. Overall, we included 1,438 respondents (N = 1,438). About 55% were females. The mean age was 46 (M = 45.65, SD = 14.08). Other baseline characteristics of the final sample are given in S1 File.

### Experimental design

The experimental design was approved by the Council of Peers at Ronald F. Inglehart Laboratory for Comparative Social Research (№ER-2020-02). It is confirmed that the proposed research project conforms to ethical standards in modern social sciences. The design was similar to Study 1, with a 2 (risk severity: high vs. low) X 2 (object at risk: individual losses vs. losses to others) factorial design. The data collection started on November 13, 2020 and finished on November 19, 2020. To ensure the effectiveness of randomization we checked for the covariate balance and put these results in S2 File. The p-values of joint orthogonality tests indicate that the group differences are insignificant, except for higher education, in the manipulation of the object at risk factor (for the details of statistical analysis, see S2 File). The CONSORT flow diagram is provided in S3 File.

### Materials

Since this was the second wave of COVID-19 pandemic we changed the wording of vignettes. We excluded some basic information which was quite new at the beginning of the COVID-19 pandemic (e.g., about the pandemic overall, spread of the issue, incubation period), but was common knowledge at the beginning of the second wave—10 months after the pandemic was declared. As a result, the number of words has been substantially decreased compared to Experiment 1. The length of text varied from 79 to 117 words. Moreover, to the moment of the second Experiment, the statistics and media coverage had changed since the beginning of the COVID-19 pandemic. Previous research has shown that the proportion of news frames has changed during different time periods depending on if it was pre-crisis, lockdown or recovery period of COVID-19 pandemic [74–76]. Following the idea of agenda-setting effects [77] we have updated some relevant and excluded some outdated information in order to make vignettes more habitual for respondents. We described the scale of the issue / recovery, medical treatment, and the advice on how to protect yourself / take care of others. The statistics and information in the vignettes were taken from Russian Federal Service for Surveillance on Consumer Rights Protection and Human Wellbeing, and Russian media sources in November 2020. Vignettes can be found in Table 5. We used the same manipulation checks. Further details of manipulation checks are given in S8 File.

Table 5. Experimental vignettes used in Experiment 2.

| | Individual Losses | Losses to Others |
|---|---|---|
| Low Risks | Although there is an outbreak of a new coronavirus, the World Health Organization recognizes that for most people, the risk of being infected is very small. | Although there is an outbreak of a new coronavirus, the World Health Organization recognizes that for most people, the risk of being infected is very small. |
| | In Russia, 98 out of 100 people recover—statistics similar to the seasonal flu. | In Russia, 98 out of 100 people recover—statistics similar to the seasonal flu. |
| | There are medications and a vaccine for the new coronavirus. | There are medications and a vaccine for the new coronavirus. |
| | In most cases, no specific medical treatment is required to recover from the coronavirus. | In most cases, no specific medical treatment is required to recover from the coronavirus. |
| | Despite the fact that the illness is most often mild, some measures are taken at the state level to prevent outbreaks of recurrent infection. At the same time, a great responsibility lies with each of us. The World Health Organization advises to take a number of measures to protect your own health. | Despite the fact that the illness is most often mild, some measures are taken at the state level to prevent outbreaks of recurrent infection. At the same time, a great responsibility lies with each of us. The responsibility not only to take care of oneself, but also of the health of other people. We are responsible for saving other lives. During a pandemic, the World Health Organization advises to take a number of measures not only to protect your health, but also the health of others. |
| High Risks | The number of infected with COVID-19 has reached almost 40 million people. More than 1 million people have died from the new coronavirus. The global coronavirus situation remains very tense. | The number of infected with COVID-19 has reached almost 40 million people. More than 1 million people have died from the new coronavirus. The global coronavirus situation remains very tense. |
| | Recently, there has been a rapid increase in the number of infected. As a result, many countries impose new restrictions and declare a second wave of new coronavirus. | Recently, there has been a rapid increase in the number of infected. As a result, many countries impose new restrictions and declare a second wave of new coronavirus. |
| | At the state level, some measures were taken to prevent outbreaks of recurrent infection, however a great responsibility lies with each of us. During a pandemic, the World Health Organization advises to take a number of measures to protect your own health. | At the state level, some measures were taken to prevent outbreaks of recurrent infection, however a great responsibility lies with each of us. The responsibility not only to take care of oneself, but also of the health of other people. We are responsible for saving other lives. During a pandemic, the World Health Organization advises to take a number of measures not only to protect your health, but also the health of others. |

## Outcome measures and covariates

We had the same three dependent variables used in Experiment 1. Descriptive statistics of the dependent variable measures and pre-treatment covariates, which are described in S4 File, used in the second experiment are given in Table 6. However, we have slightly changed the list of restrictive government policies to prevent the further spread of COVID-19, since most of the measures used in the first study were no longer relevant to the context. Similar to the first experiment, we added up the points from respondents' answers and used the resulting sum as the measure of participants' support for restrictive government policy (Cronbach's alpha, $\alpha = 0.91$; [0.90; 0.92]). The list of 10 policies included in Experiment 2 is given in S5 File. We used the 21-item short version of Scwartz's portrait values questionnaire [60] to measure prosocial values and test H2a. The focus of our experimental study was the self-transcendence direction, in particular, benevolence and universalism. The value scores were centered to measure the priority given to each of the value types as suggested by Schwartz [78].

**Table 6. Descriptive statistics of dependent variables and pre-treatment covariates.**

| Variable | N | M / % | SD | Min | Max |
|---|---|---|---|---|---|
| *Dependent Variables*: | | | | | |
| Willingness to sacrifice rights | 1438 | 2.96 | 1.14 | 1 | 5 |
| Support for restrictive government policy | 1438 | 34.61 | 9.49 | 10 | 50 |
| Support for criminal liability for quarantine violation | 1438 | 2.32 | 1.29 | 1 | 5 |
| *Moderating Variables*: | | | | | |
| Schwartz's values: Benevolence | 1426 | 0.34 | 0.83 | -3.00 | 3.10 |
| Schwartz's values: Universalism | 1426 | 0.57 | 0.70 | -2.09 | 3.10 |
| *Control Variables*: | | | | | |
| Age | 1438 | 45.65 | 14.08 | 18 | 82 |
| Female* | 1438 | 55% | -- | -- | -- |
| Higher education | 1438 | 42% | -- | -- | -- |
| Take measures to prevent COVID-19 spread | 1438 | 0.77 | 1.07 | 0 | 4 |
| Afraid of getting sick with COVID-19 | 1438 | 4.88 | 1.57 | 1 | 7 |
| Scale of COVID-19 in Russia | 1438 | 3.20 | 1.30 | 1 | 5 |
| Personal health evaluation | 1435 | 2.62 | 0.78 | 1 | 5 |
| Attitudes to the government first-wave policy | 1438 | 15.72 | 5.80 | 6 | 30 |
| Watching pro-government news | 1420 | 3.68 | 2.07 | 1 | 6 |

*Note*: Dummy variables are marked with an asterisk.

## Results

Group means comparisons are summarized in Table 7 and Fig 2. ANOVA showed statistically significant differences between four treatment conditions in the willingness to sacrifice rights ($F(3, 1434) = 2.94$, $p < 0.05$, see Table 7, Fig 2), but no differences in the support for restrictive government policies ($F(3, 1434) = 0.61$, $p = 0.608$) and criminal liability for quarantine violation ($F(3, 1434) = 0.54$, $p = 0.655$). Hence, the effect of risk framing was proven to be statistically significant in 1 of 3 measures of the dependent variable. Pairwise comparisons of experimental conditions and be found in S6 File.

Compared to Experiment 1, in Experiment 2 we found support of H1 for one measure of the dependent variable only. T-tests showed significant differences in the willingness to sacrifice rights between the low-risk (M = 2.88, SD = 1.13) and high-risk (M = 3.04, SD = 1.15) conditions ($t(1436) = -2.631$, $p < 0.01$). Hence, the ATE of risk severity on the willingness to sacrifice rights is 0.16 ([0.04; 0.28]) on a 5-point scale, Cohen's $d = 0.14$. However, no other significant differences—either in the support for restrictive government policies ($t(1436) =$

**Table 7. Group means of experimental conditions in a completely randomized 2x2 factorial design.**

| Variable | Individual losses | | Losses to others | | *F* |
|---|---|---|---|---|---|
| | Low-risk | High-risk | Low-risk | High-risk | |
| Willingness to sacrifice rights | 2.89 (1.17) | 2.98 (1.19) | 2.87 (1.10) | 3.09 (1.10) | 2.94* |
| Support for restrictive policy | 34.33 (9.79) | 35.04 (9.48) | 34.22 (9.53) | 34.84 (9.18) | 0.61 |
| Support for criminal liability | 2.25 (1.30) | 2.35 (1.30) | 2.36 (1.31) | 2.31 (1.25) | 0.54 |
| N | 362 | 356 | 346 | 374 | |

*Note*: Group means and standard deviations (in brackets) are given in the table. F-statistics are provided in the last column. Significance levels are at

*p<0.05

**p<0.01; ***p<0.001. All tests are two-tailed.

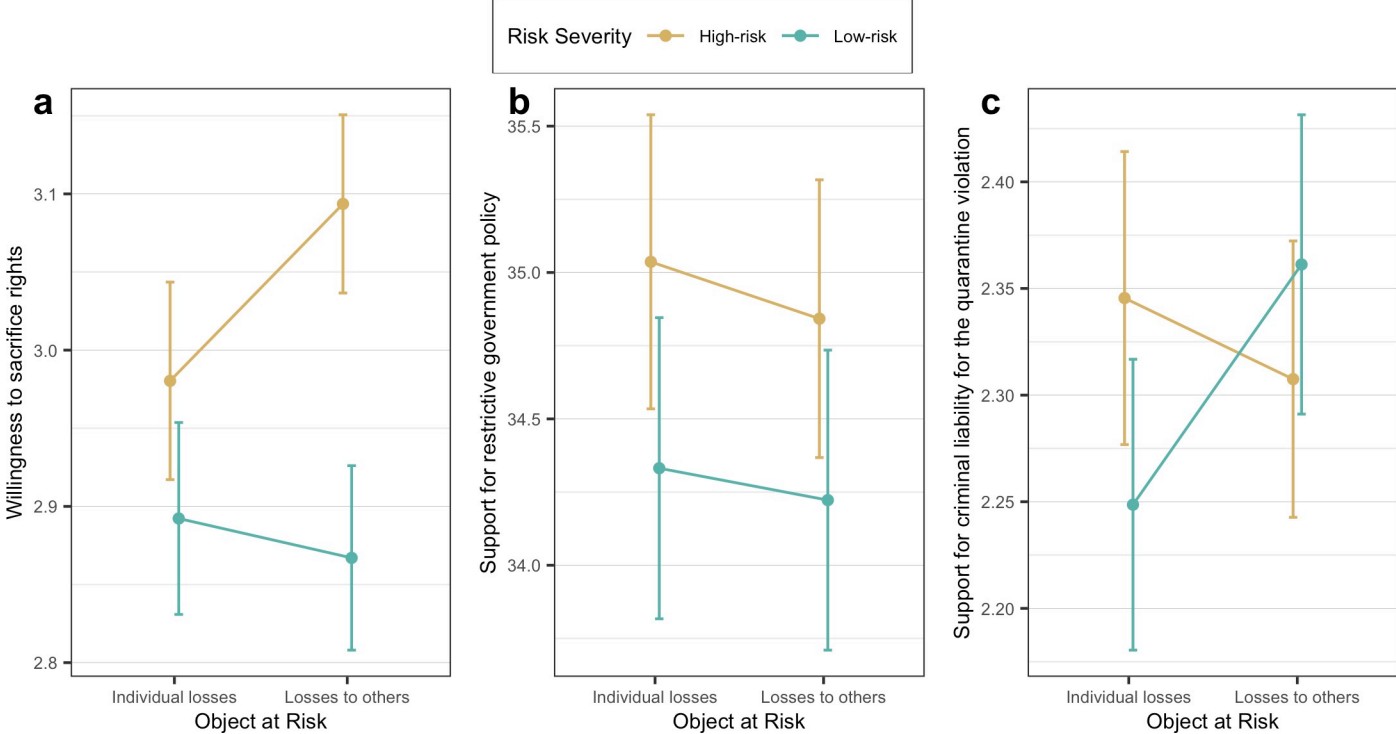

**Fig 2.** From left to right, the group means with 95% error bars in a) the willingness to sacrifice rights, b) the support for restrictive government policy, and c) the support for criminal liability for quarantine violation.

-1.317, $p = 0.188$) or in the support for criminal liability for the quarantine violation (t(1436) = -0.329, $p = 0.742$)—were found in the two other measures.

Similarly to Experiment 1 no statistically significant differences—not in the willingness to sacrifice rights (t(1436) = -0.809, $p = 0.418$), not in the support for restrictive government policies (t(1436) = 0.273, $p = 0.785$), not in the support for criminal liability for the quarantine violation (t(1436) = -0.540, $p = 0.589$)—were found between 'individual losses' and 'losses to others' conditions. So, again there are no grounds for accepting the second hypothesis. We conclude that the object at risk which indicates losses to others does not have a stronger effect on the support for restrictive government policy compared to the personal losses framing.

OLS regression models (see Table 8; see also S7 File) confirm the results of t-tests; the influence of risk severity on the willingness to sacrifice rights is the only main effect which is statistically significant ($\beta = 0.157$, $p < 0.01$). We also estimated interaction effects between main experimental factors. Models 2, 4, and 6 in Table 8 demonstrate that there are no statistically significant interaction effects between risk severity factor and object at risk, neither on willingness to sacrifice rights ($\beta = 0.138$, $p = 0.250$), nor on the support for restrictive government policy ($\beta = -0.085$, $p = 0.932$), nor on the support for criminal liability for the quarantine violation ($\beta = -0.151$, $p = 0.268$).

Now we proceed with the empirical test of the hypothesis H2a to see if the effect of the 'object at risk' framing on the support for restrictive government policy is different for the individuals with strong prosocial attitudes. In OLS models (see Table 9) we found a statistically significant interaction effect of the 'losses to others' framing and the Schwartz's value 'benevolence', which is defined by the preservation and strengthening of others' wellbeing [74], in willingness to sacrifice rights ($\beta = 0.158$, $p < 0.05$, Table 9). At 10% significance level, we also

**Table 8. OLS regression models estimates of main effects and interactions; pre-treatment covariates as controls are not included.**

| | Dependent variable: | | | | | |
|---|---|---|---|---|---|---|
| | Willingness to sacrifice rights | | Support for restrictive government policy | | Support for criminal liability | |
| | (1) | (2) | (3) | (4) | (5) | (6) |
| Intercept | 2.858*** | 2.892*** | 34.353*** | 34.331*** | 2.286*** | 2.249*** |
| | (0.052) | (0.060) | (0.432) | (0.499) | (0.059) | (0.068) |
| *Factor of Risk Severity*: High-Risk | 0.157** | 0.088 | 0.662 | 0.705 | 0.022 | 0.097 |
| | (0.060) | (0.085) | (0.501) | (0.708) | (0.068) | (0.096) |
| *Factor of Risk Target*: Losses to Others | 0.045 | -0.025 | -0.152 | -0.109 | 0.036 | 0.113 |
| | (0.060) | (0.086) | (0.501) | (0.714) | (0.068) | (0.097) |
| High-Risk X Losses to Others | | 0.138 | | -0.085 | | -0.151 |
| | | (0.120) | | (1.002) | | (0.136) |
| N | 1,438 | 1,438 | 1,438 | 1,438 | 1,438 | 1,438 |
| Adjusted $R^2$ | 0.004 | 0.004 | -0.0001 | -0.001 | -0.001 | -0.001 |
| F Statistic | 3.743* | 2.936* | 0.912 | 0.610 | 0.196 | 0.540 |

*Note*: Unstandardized beta coefficients are given in the table. Standard errors are in parentheses. Significance levels are at

†$p<0.1$

*$p<0.05$

**$p<0.01$

***$p<0.001$.

All tests are two-tailed.

found that there is a statistically significant interaction effect of the 'losses to others' framing and the Schwartz's value 'universalism'—understanding, appreciation, tolerance, and protection for the welfare of all people—on support for restrictive government policies (β = 1.374, $p < 0.1$). There were no statistically significant moderating effects of either 'benevolence' (β = 0.031, $p = 0.576$) or 'universalism' (β = -0.09, $p = 0.392$) on the support for criminal liability for quarantine violation.

Finally, similar to the results of Experiment 1, we found that the participants who were exposed to the 'low-risk' framing ($M = 2.76$, $SD = 0.65$) were less convinced of the credibility of the information ($t(1436) = -4.936$, $p < 0.001$) than those who were in the 'high-risk group' ($M = 2.92$, $SD = 0.64$); see S8 File for more details. We found that the CATEs of risk severity framing are also higher for those who perceive the treatment information as credible (see S8 File). In other words, Experiment 2 demonstrates empirical evidence of H1 for all three measures of attitudes towards restrictive government policy, but this effect of high vs. low risks is observed only for those who perceive information as credible (see main effects and interactions in Table 10). On the contrary, in Experiment 1 we found that the framing effect was consistent among all respondents, though it was also conditioned by perceived information credibility.

## Discussion

There are three major findings in this study. First, focusing on higher risks has a positive effect on the support for the government restrictive policy. We found some evidence for the first hypothesis H1, i.e., that high-risk framing caused a higher willingness to sacrifice rights, support for government restrictive measures, and criminal liability. That is in line with the literature on both risk perception of infections and the perception of societal risks which can be a threat to society and social order overall [24–26]. This is also in consistency with health-related behavior theories (e.g., protection motivation theory, health belief model) and the research

**Table 9. OLS regression models estimates of CATEs using values as moderators; pre-treatment covariates as controls are not included.**

| | Dependent variable: | | | | | |
|---|---|---|---|---|---|---|
| | Willingness to sacrifice rights | | Support for restrictive government policy | | Support for criminal liability | |
| | (1) | (2) | (3) | (4) | (5) | (6) |
| Intercept | 2.858*** | 2.810*** | 34.337*** | 34.562*** | 2.273*** | 2.338*** |
| | (0.054) | (0.063) | (0.456) | (0.530) | (0.062) | (0.072) |
| *Factor of Risk Severity*: High-Risk | 0.158** | 0.160** | 0.729 | 0.715 | 0.025 | 0.025 |
| | (0.060) | (0.060) | (0.504) | (0.503) | (0.068) | (0.068) |
| *Factor of Risk Target*: Losses to Others | -0.008 | 0.003 | -0.191 | -0.920 | 0.050 | -0.016 |
| | (0.065) | (0.078) | (0.543) | (0.650) | (0.074) | (0.088) |
| Schwartz's values: Benevolence | -0.006 | | -0.049 | | 0.031 | |
| | (0.050) | | (0.422) | | (0.057) | |
| Losses to Others X Benevolence | 0.158* | | 0.078 | | -0.046 | |
| | (0.072) | | (0.606) | | (0.082) | |
| Schwartz's values: Universalism | | 0.076 | | -0.389 | | -0.090 |
| | | (0.060) | | (0.503) | | (0.068) |
| Losses to Others X Universalism | | 0.085 | | 1.347† | | 0.084 |
| | | (0.086) | | (0.719) | | (0.098) |
| N | 1,426 | 1,426 | 1,426 | 1,426 | 1,426 | 1,426 |
| Adjusted $R^2$ | 0.009 | 0.009 | -0.001 | 0.002 | -0.002 | -0.001 |
| F Statistic | 4.104** | 4.086** | 0.550 | 1.566 | 0.189 | 0.537 |

*Note*: Unstandardized beta coefficients are given in the table. Standard errors are in parentheses. Significance levels are at

†$p < 0.1$

*$p < 0.05$

**$p < 0.01$

***$p < 0.001$.

All tests are two-tailed.

**Table 10. Conditional average treatment effects of high-risk vs. low-risk groups.**

| | Dependent variable: | | |
|---|---|---|---|
| | Willingness to sacrifice rights | Support for restrictive government policy | Support for criminal liability |
| Intercept | 1.893*** | 28.779*** | 1.599*** |
| | (0.223) | (1.758) | (0.227) |
| Risk Severity: High-Risk | -0.984** | -10.567*** | -0.866** |
| | (0.302) | (2.512) | (0.301) |
| Perceived credibility | 0.358*** | 1.995*** | 0.256** |
| | (0.077) | (0.606) | (0.081) |
| High-Risk X Perceived credibility | 0.370*** | 3.724*** | 0.289** |
| | (0.103) | (0.840) | (0.107) |
| N | 1,438 | 1,438 | 1,438 |
| Adjusted $R^2$ | 0.108 | 0.084 | 0.044 |
| F Statistic (df = 3; 1434) | 58.968*** | 45.160*** | 22.882*** |

*Note*: Unstandardized beta coefficients are given in the table. Standard errors are in parentheses. Significance levels are at

*$p < 0.05$

**$p < 0.01$

***$p < 0.001$.

All tests are two-tailed.

results which show that those who evaluate health risks as high are more willing to comply with self-protective measures [79]. Our finding also mirrors earlier studies of pandemic impact on social attitudes and behavior. The Ebola outbreak, for instance, was found to produce a stronger support for restrictive policies [80].

At the same time, it should be noted that in terms of effect sizes the differences were small and not always statistically significant. Cohen's *d* was up to 0.27 in Experiment 1 and up to 0.16 in Experiment 2. Overall, the participants in both experiments strongly approved a number of restrictive policies. Small effect sizes might result from the fact that the overall level of anxiety and emotional fear ('affect heuristic' [21]) of COVID-19 is very high. Significantly more respondents showed a lower level of credibility to the vignettes in the low-risk condition in both experiments. The effect sizes were higher among those who found the information in the vignettes more credible: Cohen's *d* was up to 0.34 in Experiment 1 and up to 0.19 in Experiment 2. This result was anticipated, since previous research has shown that individuals seem to perceive lower risk estimates as less credible and consider such information as less trustworthy, especially in health communication [81]. In addition, our result is consistent with the so-called negativity bias effect in processing information [82]. The high-risk condition can be also called a strong frame as it is more compelling for people [33]. That is also elicited by the substantial overestimation of the COVID-19 mortality and infection rates by the respondents in both experiments. The mean mortality rate was evaluated as 11% in Experiment 1 and 35% in Experiment 2. The mean infection rate was evaluated as 31% in Experiment 1 and 56% in Experiment 2. Due to the availability heuristic, when the information about the number of deaths and newly infected people is reported on a daily basis, this increased both infection and mortality rates. This is in accordance with the literature that shows an increase in risk evaluation if the issue is salient for people and if mass media reports the risks on a regular basis [23]. Indeed, the higher individuals evaluated the infection and fatality rates, the more they supported government restrictive policy.

Second, focusing on the losses for others did not produce a stronger support for the restrictive policy compared to focusing on personal losses. This is in line with the papers which found no difference between self-focused and prosocial framing during the COVID-19 pandemic [47, 48, 53, 54]. We found no evidence that prosocial responsibility acts like a Trojan horse for willingness to sacrifice rights and an acceptance of privacy violation [83]. This effect is in line with the idea that people are less inclined to sacrifice for others in a state of uncertainty [84], but contradicts the opposing point of view that people tend to sacrifice if they are exposed to worst-case scenarios [85]. Alongside health and death issues, pandemics might also impact a conservative shift and security demands for oneself [86].

Third, though focusing on the losses for others did not produce a stronger support for restrictive policy, we found a positive moderation effect of such prosocial values as universalism and benevolence. We found that those with prosocial values had a stronger positive effect in the "losses for others" frame and were more willing to support restrictive policy when others were included. This is in line with the prediction of Wolf et al. paper though they showed no empirical evidence of the claim [63]. It seems that proclaiming the importance of being 'supportive' and 'careful' towards others by WHO, during the pandemic, may increase the support for restrictive policy by some individuals whose values are activated when others are included as those who can be harmed. This is in accordance with the literature that links prosocial values with prosocial risk taking in which there may be a risk for others [45]. This is also in line with the literature on the effect of frames which showed a moderator effect of values. Chong and Druckman [33] emphasize that people's preferences are a function of personal values and the strength of competing frames on the issue. As a result, some frames should be in consistency with personal values to have an effect on attitudes in a competitive environment.

The effects found in our experimental study reveal both positive and negative aspects in risk communication during the pandemic, which may have a great and long-term impact on trust, attitudes, and behavior. The major positive aspect is the efficiency of risk communication for the awareness of risks and its recognition. Higher perceived risks result in a more risk-averse behavior and higher willingness to undertake protective measures [87].

However, at the same time there might be some negative consequences of this risk communication. First, higher risk perception can undermine social trust, political trust, and trust in scientific experts, if protective measures bring negative consequences to the population or the government is not able to handle the issue [88]. According to the protection motivation theory and health belief model, the perceived effectiveness of recommended measures has an effect on the willingness to follow precautionary actions. If perceived effectiveness is quite low, this can bring a decrease of political trust in government similar to what has happened in Europe after the H1N1 pandemic in 2009 [89]. It was also shown that forced social distance and the social changes caused by the Spanish flu had a long-term negative effect on social trust [90]. Second, a greater loyalty to discretionary power of the executives may seem similar to Slovic's outlook on the consequences of nuclear risks for democracy [91]. While risk communication produces a high-risk perception of COVID-19, it can be efficient in promoting anti-democratic ideas and messages or the denigration of minorities. Similarly, while including others as those who can be harmed, it can be efficient in promoting support for anti-democratic policies among those who have prosocial values. Thus, there might be growing support for restrictive policy worldwide.

The recent papers published in *the Lancet*, which argue that 'we need to pay attention to how authoritarian forces shape our frame of mind' during the pandemic [92], are a wake-up call for risk communication research. Nevertheless, risk framing and its effects on support for restrictive government policy is critical not just to understand the coercive apparatus of authoritarian government, but to evaluate countries' ability to conduct risk communication in shaping people's risk perceptions and instructing them to adopt certain preventive measures, such as social distancing and self-isolation. Overall, we stress the importance of the further exploration of risk communication during the COVID-19 pandemic in different cultures and different population groups, as this has tremendous long-term consequences for all countries.

There are several limitations in this study. First, we cannot make generalizations about Russia since a non-probability sample was used in both experiments. Second, we cannot fully extrapolate our findings to other countries. Both findings can be culturally specific [37]. Russia is a developing country with a certain socio-cultural background and cultural values. Cross-cultural studies should be conducted to explore the differences in risk perception, and the effect of different risk framing on risk perception and the support for restrictive policy. Third, the wording of the vignettes was different in our experiments which make our conclusion about the effect sizes limited, since the change in effect sizes can be due to different wording, but also due to different time points and different survey populations. The effect sizes can also vary in further experiments depending on COVID-19 risk dynamics and media coverage of COVID-19.

In spite of the limitations, our findings confirm the great political importance of risk communication and risk literacy in the time of the pandemic. Gigerenzer [93] shows that risk education and the improvement of risk literacy with regards to staying healthy, should be the focus of institutions, as acting politicians can make ill-advised decisions in the time of crises and pandemics, when long-term consequences of protective measures cannot be examined.

## Supporting information

**S1 File. Eligibility and baseline sample characteristics.**
(DOCX)

**S2 File. Randomization checks.**
(DOCX)

**S3 File. CONSORT flow diagrams.**
(DOCX)

**S4 File. Selection and measurement of pre-treatment covariates.**
(DOCX)

**S5 File. DV measurement: support for restrictive government policy.**
(DOCX)

**S6 File. ANOVA post hoc analyses.**
(DOCX)

**S7 File. Results of OLS models.**
(DOCX)

**S8 File. Manipulation checks.**
(DOCX)

**S9 File. References used in supporting information.**
(DOCX)

## Acknowledgments

This paper is dedicated to our mothers. We thank them for all their endless love they gave to us. We miss you a lot. May you all rest in peace.

## Author Contributions

**Conceptualization:** Kirill Chmel, Aigul Klimova, Nikita Savin.

**Data curation:** Kirill Chmel, Aigul Klimova.

**Formal analysis:** Kirill Chmel, Aigul Klimova.

**Investigation:** Kirill Chmel, Aigul Klimova, Nikita Savin.

**Methodology:** Kirill Chmel, Aigul Klimova, Nikita Savin.

**Project administration:** Kirill Chmel.

**Software:** Kirill Chmel, Aigul Klimova.

**Validation:** Kirill Chmel, Aigul Klimova.

**Visualization:** Kirill Chmel.

**Writing – original draft:** Kirill Chmel, Aigul Klimova, Nikita Savin.

**Writing – review & editing:** Kirill Chmel, Aigul Klimova, Nikita Savin.

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
