## [Decision Letter · Decision Letter 0]

17 May 2021

PONE-D-21-10676

The effect of risk framing on support for restrictive government policy regarding the COVID-19 outbreak

PLOS ONE

Dear Dr. Chmel,

Thank you for submitting your manuscript to PLOS ONE. After careful consideration, we feel that it has merit but does not fully meet PLOS ONE’s publication criteria as it currently stands. Therefore, we invite you to submit a revised version of the manuscript that addresses the points raised during the review process.

Please find below the reviewers' comments, as well as those of mine.

We look forward to receiving your revised manuscript.

Kind regards,

Valerio Capraro

Academic Editor

PLOS ONE

Journal Requirements:

Additional Editor Comments:

I have now collected three reviews from three experts in the field. All reviewers think that this paper has the potential to make a valuable contribution to the literature, but they suggest several improvements before publication. I am myself familiar with the topic of this manuscript, and I agree with the reviewers. Therefore, I would like to invite you to revise your work for Plos One following their suggestions. On top of their comments, I would like to add one more comment from my own reading. I was surprised not to see a discussion of the literature regarding the effect of proself and prosocial messages on pandemic response, which is very relevant to your research, as you also test the effect of framing a message as a loss to self vs others. There are several works that have explored the effect of proself and prosocial messages; these are the ones I know: Banker & Park (2020), Bilancini et al (2020), Capraro & Barcelo (2020), Heffner et al (2020), Lunn et al (2020), Pfattheicher et al (2020) - but it is possible that there are others (please double check). This line of work is very relevant and I think it should be discussed.

I am looking forward for the revision.

References

Banker, S., & Park, J. (2020). Evaluating prosocial COVID-19 messaging frames: Evidence from a field study on Facebook. Judgment and Decision Making, 15(6), 1037-1043.

Bilancini E, Boncinelli L, Capraro V, Celadin T, Di Paolo R (2020) The effect of norm-based messages on reading and understanding COVID-19 pandemic response governmental rules. Journal of Behavioral Economics for Policy 4, Special Issue 1, 45-55.

Capraro, V., & Barcelo, H. (2020). The effect of messaging and gender on intentions to wear a face covering to slow down COVID-19 transmission. Journal of Behavioral Economics for Policy, 4, Special Issue 2, 45-55.

Heffner, J., Vives, M. L., & FeldmanHall, O. (2020). Emotional responses to prosocial messages increase willingness to self-isolate during the COVID-19 pandemic. Personality and Individual Differences, 170, 110420.

Lunn, P. D., Timmons, S., Barjaková, M., Belton, C. A., Julienne, H., & Lavin, C. (2020). Motivating social distancing during the Covid-19 pandemic: An online experiment. Social Science & Medicine, 113478.

Pfattheicher, S., Nockur, L., Böhm, R., Sassenrath, C., & Petersen, M. B. (2020). The emotional path to action: Empathy promotes physical distancing during the COVID-19 pandemic. Psychological Science, 31, 1363-1373.

Reviewers' comments:

Reviewer's Responses to Questions

**Comments to the Author**

1. Is the manuscript technically sound, and do the data support the conclusions?

Reviewer #1: Yes

Reviewer #2: Partly

Reviewer #3: Yes

2. Has the statistical analysis been performed appropriately and rigorously? 

Reviewer #1: Yes

Reviewer #2: Yes

Reviewer #3: Yes

3. Have the authors made all data underlying the findings in their manuscript fully available?

Reviewer #1: Yes

Reviewer #2: Yes

Reviewer #3: Yes

4. Is the manuscript presented in an intelligible fashion and written in standard English?

Reviewer #1: Yes

Reviewer #2: Yes

Reviewer #3: Yes

5. Review Comments to the Author

Reviewer #1: This study tested the effect of severity of risk description (low vs. high) and object at risk (self vs. others) on people’s attitude toward government restrictive policies. With two experiments at two different stages during the covid pandemic, the study found the severity of risk was consistently significant. On the other hand, the effect of object at risk was moderated by prosocial values.

This study has multiple positive features. Its topic has an important practical implication. The sample size was relatively large. The research questions were tested twice in different stages. The main text was concise but still informative (I like the comprehensive information in the appendix). Overall, I believe the conclusion of this study was supported by its methodology and analyses.

I have a few questions and I hope the authors can clarify.

First, this study tested issue framing and the key manipulation was severity of the risk. Based on my reading of your materials, it seems higher risk condition was associated with more significant losses. As a result, you did find that higher risk led to more positive ratings on the government restrictive policies. Hence, it appears more losses might cause people to take actions to mitigate the spread of the virus (i.e., function of loss aversion). Interestingly, some studies with gain-loss framing found mixed results on the effect of loss aversion (for your reference see a recent study https://psyarxiv.com/4wc5d/ This study also tested risk attitude and political ideology as relevant to your study). While your study did not directly test gain-loss framing, can you still discuss some implications on loss aversion and gain-loss framing based on your findings?

Second, you adopted a typical 2*2 design and I was wondering why you used t-test repeatedly instead of running an anova with post hoc t-tests? Repeat t-test might inflate type I error. Relatedly (I was confused here so hope you can clarify), for your anvoa test, what was the independent variable? For example, you stated “As expected, we found statistically significant differences in the support for restrictivegovernment policy (F(3, 762) = 4.68, p < 0.01, see Table 2, Fig 1) and the support for criminal liability for the quarantine violation (F(3, 762) = 2.72, p < 0.05).” In other words, for these F-tests, which compared to which?

In the main text you stated there was no interaction. However, looking at Figures 1b, 1c, 2a, and 2c, it seems there are interactions between risk severity and object at risk. Did you test these interactions with either anova or OLS?

I like your variable of watching pro-government news. Since two of your dependent variables dealt with government policies, was there an interaction between risk severity and watching pro-government news?

Reviewer #2: Thank you for the opportunity to review this interesting paper about the effect of risk framing on the support for governmental COVID-19 measures in Russia. This study provides some useful tests for the influence of a high risk versus low risk frame as well as the frame of self-protection compared to the responsibility to protect others on the acceptance of coronavirus protection policies. With help of two different survey experiments, the authors show that by framing the coronavirus as risky and dangerous, the willingness to sacrifice some personal rights for the greater good can be increased. In the course of this, they also show the importance of how credible the information is perceived to be. They do not find support for their hypothesis that framing others to be at risk increases the acceptance of restrictive policy measures. However, they mention evidence that people with prosocial attitudes react to the solidarity treatment, and claim to be more willing to sacrifice own rights for the protection of others.

Overall, this survey experiment in Russia provides some interesting and new insights about the acceptance of COVID-19 measures and to what extent the coronavirus support is influenced by different kinds of risk framing. In general, procedure and results are nicely described, there is a great quantity of different analyses and checks, and the study is written very comprehensible. However, there are some aspects that should respectively could be improved - in particular concerning the provision of analyses results and the discussion of the procedure and the results. Since most of the following comments only entail minor revisions, their implementation should be possible without major difficulties.

First, I would like to state one major issue concerning the test of hypothesis 2a that needs to be addressed.

a) Whereas there is plenty of information (such as descriptive tables, t-tests, and randomization and manipulation checks) in the whole paper and in the supporting information, there is a lack of important information concerning the tests of hypothesis 2a. First of all, for better orientation in the paper, it would be helpful, if it is directly mentioned that hypothesis 2a is studied and discussed now (page 17 and/or in the discussion). More important than this is that the analyses and results of H2a, which lead to the conclusions, need to be sufficiently presented. Although the authors mention the OLS models testing H2a, they do not present enough of the model and results (neither in the text nor in the supporting information). For instance, it is not clear if controls had been included and if so which ones. Also, as a reader I would like to look at the main effects, too, and be able to see the whole picture of main effects and the interaction (which can be enlightening and somewhat surprising as in the case of the credibility interaction in Table 5). Because of this, and since the authors promote the interaction of prosocial persons and the willingness to sacrifice own rights for others as a major finding of the study, I think, they have to present the results of the complete OLS models somewhere (at least in the supporting information).

b) Additionally, I missed the presentation of the moderating Schwartz values "Benevolence" and "Universalism" in the supporting information at the randomization checks (chapter B2). However, instead of the moderating variables there is another Schwartz value presented ("Security"), which is not mentioned anywhere else. It would be helpful to be more consistent and more precise about which Schwartz values had been used for which analyses, and why.

c) In experiment 1, I do not fully understand why hypothesis 2a cannot be tested due to too little variation in the moderating variables. What does this mean here and where can this information be found? For instance, the Schwartz values are not presented in the descriptive table here. Does this mean that these variables had not been conducted in the first experiment and therefore included in the second study to test H2a? Alternatively, had all students (more or less) the same attitudes? If there are further results of this topic, which had not been presented yet, it should be considered to provide them – at least in the supporting information.

Second, I want to give some comments on minor issues that could help the authors to improve their paper.

1) Since this study was carried out in Russia, I suggest adding more detailed information about the situation as well as the measures in this country already in the introduction. In the introduction, countries like Sweden and Austria are mentioned, but not Russia, where the experiments had been carried out. As a reader, I would like to know more COVID-19 in Russia. There is even some information about Russia in the cited Gallup poll. [The cited link to the poll does not work anymore. But, here is one that worked for me: https://www.gallup-international.com/fileadmin/user_upload/surveys/2020/GIA_SnapPoll_2020_COVID_Tables_final.pdf

Also, the information from the Gallup poll is from the very beginning of the COVID-19 pandemic. I would also like to see some more recent information – ideally from around the time when the second nationwide experiment took place – because it is possible that the attitudes changed from the first to the second wave. Also, the restrictive policy measures changed over, as we can see in the supporting information. However, some more detailed description in the text would be interesting and could help to develop a better understanding of the situation and the results. So, to me, the chapter "Context of the study, COVID-19 in Russia" could be extended or alternatively included in the introduction.

2) The pre-treatment covariates of both experiments are presented in Table 1 and Table 3. However, most of seem to be a little lost, since they are not mentioned or explained (such as "Scale of COVID-19 in Russia", "Watching pro-government new" or "Take measures to prevent COVID-19 spread"). I see that they are not of main interest in this study, but some further information would be helpful. Why had just these pre-treatment covariates been included, and considered being important for this study? At least in the supporting information there could be further explanation about the purpose and consequences of these variables and their relevance in terms of the two factors of the survey experiment.

3) In the second experiment, the wording of the vignettes – and therefore the treatment of the study – had been changed. The first question that arises here is: Why? Had there been some issues with the vignettes in the first experiment? At the first glance, manipulation checks (supporting information G1) do not suggest to change the vignettes. So, what were the reasons behind this decision? Second, is it possible, that the change of the vignettes affected the results or are the some arguments why this is not likely or why the changing was even necessary? I see, that the results of H1 are somewhat ambivalent (there is less support for H1 in the second experiment). There could be various possible reasons for the differences in the results between experiment 1 and 2. They had been carried out at different points of time (first vs. second wave). The population/sample is not the same (student sample vs. nationwide internet users). The treatment had been changed. I think, that these decisions need to be justified, and their impact and the arising questions should be discussed.

4) Discussion, page 20: Unfortunately, I cannot follow how the authors derive that the support for H1 shows that there is more risk averse behavior in the high-risk framing. As far as I understood the study, it was more an effect of willingness to sacrifice rights and support of restrictive measures than actual risk averse behavior (such as staying at home or reducing meetings with others). So, I would appreciate if the authors would think about their wording here, or could help me understand this conclusion better by some further explanation.

5) There seems to be an issue with the tables in the results part of the second experiment. There are two Table 3 (page 15 and page 16). Table 4 is mentioned in the text on page 18, but the content of this Table 4 is the same as in Table 3 on page 16. So, reading the text, I guess that Table 3 of page 16 is actually Table 4. Table 4 from page 18 can be dropped. Table 5 contains the right information about the credibility interaction. Also, please be aware t hat basically the same section is accidentally included twice ["In other words, the second experiment demonstrates empirical evidence of the H1 for all three measures of attitudes towards restrictive government policy, but the framing effect of high vs. low risks is observed only for those who perceive information as credible (see main effects and interactions in Table 4/5). On the contrary, in the first study we found that the framing effect was consistent among all respondents, though it was also conditioned by the perceived information credibility."; this section can be found on page 18 before "Table 4" and on page 19 after the end of Table 5.]

6) The citation of Trumbo and McComas (2003) is formally not in harmony with the remaining citations. The resource is mentioned on page 20 and in the supporting information. Bur this paper is not listed as a resource; hence, it should be added to the references section.

7) Looking at Figure 2 and reading that there is support for hypothesis 1, it seems that the descriptive statistics of high-risk and low-risk are accidentally interchanged ["… willingness to sacrifice rights between the high-risk (M = 2.88; SD = 1.13) and low-risk (M = 3.04; SD = 1.15) conditions … "] on page 16.

8) Finally, (though somewhat funny) the typo on page 9 – farming instead of framing – should be corrected.

Reviewer #3: Review for PONE-D-21-10676

This paper examines the effects of message framing on support for COVID-related restrictions. Overall while the theoretical contribution is not new, I do believe that the studies provide useful insights into the efficacy of message framing approaches in Russia during the pandemic. My suggestions primarily involve improving the clarity of paper.

- Because there are now a number of published and working papers on the effects of COVID-related message framing, it would help to contextualize the current work by more thoroughly summarizing the related literature.

- In addition, I appreciated the detailed web appendix. However, I feel that it would help to facilitate understanding by moving as much of those details as possible into the main paper so that the reader does not need to refer to the appendix for essential information.

- For example, it was difficult to understand the experimental design without seeing the stimuli. I would recommend moving the vignettes to the main text so that readers understand what is manipulated.

- Were the study and analyses exploratory? If so, this should be stated in the paper.

- Was there a sample size goal? How was the stopping criteria for the study determined?

- In the dependent measures, how were the scale anchors determined? Why was “willingness” measured with a ready/not ready scale? It seems that this wording confuses the acceptance of restrictive measures with the expectation/anticipation of restrictive measures. Please clarify.

- Please explain the high 51% drop-out rate with greater detail (e.g., what was the exact percent of people dropping out on the first page?), were there differences between conditions?

- Statistical details on all test should be reported. For example, I did not find details related to this statement: “The p-values of joint orthogonality tests indicate that the group differences are insignificant, except for probability of COVID-19 infection in the manipulation of the risk severity factor.”

- Please proofread the manuscript for typos. For example on page 9: “The vignettes were structured as the set of rubrics with essentially similar content, yet different farming.” P9

In general, I believe that this paper offers a nice documentation of message framing effects in the Russian context. Adding further detail and clarity to the main text will help to improve the impact of the work.

6. PLOS authors have the option to publish the peer review history of their article (what does this mean?). If published, this will include your full peer review and any attached files.

Reviewer #1: No

Reviewer #2: No

Reviewer #3: No

---

## [Author Response · Author response to Decision Letter 0]

30 Jun 2021

Dear Editorial Board,

We thank all three Reviewers and the Editor for their generous comments and suggestions on the manuscript. We have edited the manuscript to address their concerns. For convenience, we put our responses in a file entitled 'Response to Reviewers' and attach it to the submission. In this rebuttal letter we describe the changes made in response to the reviewers’ comments below. We also attach the revised manuscript and the revised version of supplementary materials with highlighted changes.

Please, let us know, if we need to clarify our response or provide it in a different format.

Thank you for your consideration of the manuscript.

Sincerely,

Authors

---

## [Decision Letter · Decision Letter 1]

22 Jul 2021

PONE-D-21-10676R1

The effect of risk framing on support for restrictive government policy regarding the COVID-19 outbreak

PLOS ONE

Dear Dr. Chmel,

Thank you for submitting your manuscript to PLOS ONE. After careful consideration, we feel that it has merit but does not fully meet PLOS ONE’s publication criteria as it currently stands. Therefore, we invite you to submit a revised version of the manuscript that addresses the points raised during the review process.

We look forward to receiving your revised manuscript.

Kind regards,

Valerio Capraro

Academic Editor

PLOS ONE

Journal Requirements:

Additional Editor Comments (if provided):

All reviewers are satisfied with the revision and suggest acceptance. I have selected minor revision in the editorial manager just because one of the reviewers noticed that one reference is missing. Please address this comment at your earliest convenience. I am looking forward to receive the final version.

Reviewers' comments:

Reviewer's Responses to Questions

**Comments to the Author**

1. If the authors have adequately addressed your comments raised in a previous round of review and you feel that this manuscript is now acceptable for publication, you may indicate that here to bypass the “Comments to the Author” section, enter your conflict of interest statement in the “Confidential to Editor” section, and submit your "Accept" recommendation.

Reviewer #1: (No Response)

Reviewer #2: All comments have been addressed

Reviewer #3: All comments have been addressed

2. Is the manuscript technically sound, and do the data support the conclusions?

Reviewer #1: (No Response)

Reviewer #2: Yes

Reviewer #3: Yes

3. Has the statistical analysis been performed appropriately and rigorously? 

Reviewer #1: (No Response)

Reviewer #2: Yes

Reviewer #3: Yes

4. Have the authors made all data underlying the findings in their manuscript fully available?

Reviewer #1: (No Response)

Reviewer #2: Yes

Reviewer #3: Yes

5. Is the manuscript presented in an intelligible fashion and written in standard English?

Reviewer #1: (No Response)

Reviewer #2: Yes

Reviewer #3: Yes

6. Review Comments to the Author

Reviewer #1: Thank you for your efforts addressing my comments. I am good with the current version. Again, I think the study makes a nice contribution to understand health behaviors in the current pandemic.

Reviewer #2: The authors have responded very well and in detail to my suggestions and questions. The revised sections are enlightening and offer interesting additions. Also, I greatly appreciate the newly created supporting material S4, which is helpful to understand measurement and purpose of the considered pre-treatment covariates. A brief note in this regard: I could not find the references cited in S4, hence, it might be helpful to add them directly at the end of the tables in S4. Overall, the paper has been significantly improved and clarified after revising. Therefore, I support and recommend publication.

Reviewer #3: The authors have adequately addressed all my comments. Thank you for the responsive revision and nice work.

7. PLOS authors have the option to publish the peer review history of their article (what does this mean?). If published, this will include your full peer review and any attached files.

Reviewer #1: No

Reviewer #2: No

Reviewer #3: No

---

## [Author Response · Author response to Decision Letter 1]

27 Jul 2021

Dear Editorial Board,

We again thank all three Reviewers and the Editor for their generous comments and suggestions on the manuscript. We have edited the manuscript to address concerns of the Reviewer #2. In this rebuttal letter we describe the changes made in response to the comments. We also attach the revised manuscript and the revised version of supplementary materials with highlighted changes.

Response to the Reviewer #2: The authors have responded very well and in detail to my suggestions and questions. The revised sections are enlightening and offer interesting additions. Also, I greatly appreciate the newly created supporting material S4, which is helpful to understand measurement and purpose of the considered pre-treatment covariates. A brief note in this regard: I could not find the references cited in S4, hence, it might be helpful to add them directly at the end of the tables in S4. Overall, the paper has been significantly improved and clarified after revising. Therefore, I support and recommend publication.

We thank the Reviewer for careful reading of the manuscript and the particular attention to the Supporting information. Thanks to this comment, we noticed that there are no references listed for the Supporting information. To address this issue, we updated the Supporting information and added one more file - S9 File – in which the references used in the Supporting information are listed.

Response to the Editor: All reviewers are satisfied with the revision and suggest acceptance. I have selected minor revision in the editorial manager just because one of the reviewers noticed that one reference is missing. Please address this comment at your earliest convenience. I am looking forward to receiving the final version.

We thank the Editor for the opportunity to revise again the manuscript. We made sure that the References section is complete and added to the Supporting information another file in which the references used in the Supporting information are listed.

Thank you for your consideration of the manuscript.

Sincerely,

Authors

---

## [Decision Letter · Decision Letter 2]

21 Sep 2021

The effect of risk framing on support for restrictive government policy regarding the COVID-19 outbreak

PONE-D-21-10676R2

Dear Dr. Kirill Chmel,

We’re pleased to inform you that your manuscript has been judged scientifically suitable for publication and will be formally accepted for publication once it meets all outstanding technical requirements.

Kind regards,

Akihiro Nishi, M.D., Dr.P.H.

Academic Editor

PLOS ONE

Additional Editor Comments (optional):

I am happy to see that the two reviewers are satisfied with the quality of the work.

Reviewers' comments:

Reviewer's Responses to Questions

**Comments to the Author**

1. If the authors have adequately addressed your comments raised in a previous round of review and you feel that this manuscript is now acceptable for publication, you may indicate that here to bypass the “Comments to the Author” section, enter your conflict of interest statement in the “Confidential to Editor” section, and submit your "Accept" recommendation.

Reviewer #1: (No Response)

Reviewer #2: All comments have been addressed

2. Is the manuscript technically sound, and do the data support the conclusions?

Reviewer #1: (No Response)

Reviewer #2: Yes

3. Has the statistical analysis been performed appropriately and rigorously? 

Reviewer #1: (No Response)

Reviewer #2: Yes

4. Have the authors made all data underlying the findings in their manuscript fully available?

Reviewer #1: (No Response)

Reviewer #2: Yes

5. Is the manuscript presented in an intelligible fashion and written in standard English?

Reviewer #1: (No Response)

Reviewer #2: Yes

6. Review Comments to the Author

Reviewer #1: I recommended accept in the last round. The minor change in the latest round did not change my decision.

Reviewer #2: Dear Authors

Thank you for including the references of the Supporting Information in an additional file (S9). All concerns have been very well addressed. Therefore, I support and recommend publication.

7. PLOS authors have the option to publish the peer review history of their article (what does this mean?). If published, this will include your full peer review and any attached files.

Reviewer #1: No

Reviewer #2: No

---

## [Editor Report · Acceptance letter]

24 Sep 2021

PONE-D-21-10676R2 

The effect of risk framing on support for restrictive government policy regarding the COVID-19 outbreak 

Dear Dr. Chmel:

I'm pleased to inform you that your manuscript has been deemed suitable for publication in PLOS ONE. Congratulations! Your manuscript is now with our production department. 

Kind regards, 

on behalf of

Dr. Akihiro Nishi 

Academic Editor

PLOS ONE